# Exploiting Conserved Quorum Sensing Signals in *Streptococcus mutans* and *Streptococcus pneumoniae*

**DOI:** 10.3390/microorganisms10122386

**Published:** 2022-11-30

**Authors:** Giulia Bernabè, Anthony Pauletto, Annj Zamuner, Leonardo Cassari, Ignazio Castagliuolo, Paola Brun, Monica Dettin

**Affiliations:** 1Department of Molecular Medicine, Section of Microbiology, University of Padova, Via A. Gabelli 63, 35127 Padova, Italy; 2Department of Industrial Engineering, University of Padova, Via Marzolo 9, 35131 Padova, Italy

**Keywords:** quorum quenching, competence-stimulating peptide, peptide synthesis, microbiota, bacteriocin

## Abstract

Bacterial species of the *Streptococcus* genera are considered either commensal bacteria or potential pathogens, according to their metabolic evolution and production of quorum sensing (QS)-controlled virulence factors. *S. mutans,* in particular, has become one of the best-studied examples of bacteria that are able to get along or cheat commensal species, even of the same genera. *S. mutans* and *S. pneumoniae* share homolog QS pathways and a competence stimulating peptide (CSP) for regulating bacteriocin production. Intriguingly, the abundance of *S. pneumoniae* and *S. mutans* alternates in complex microbial communities, thus opening the role for the fratricide communication of homolog QS systems. Since the inhibition of the QS has been proposed in treating bacterial infections, in this study, we designed and synthesized analogs of *S. pneumoniae* CSP with precise residual modifications. We reported that *S. pneumoniae* CSP analogs reduced the expression of genes involved in the QS of *S. mutans* and biofilm formation without affecting bacterial growth. The CSP analogs inhibited bacteriocin production in *S. mutans,* as reported by co-cultures with commensal bacteria of the oral cavity. The peptide CSP1AA, bearing substitutions in the residues involved in QS receptor recognition and activation, reported the most significant quorum-quenching activities. Our findings provide new insights into specific chemical drivers in the CSP sequences controlling the interconnection between *S. mutans* and *S. pneumoniae*. We think that the results reported in this study open the way for new therapeutic interventions in controlling the virulence factors in complex microbial communities such as the oral microbiota.

## 1. Introduction

Different microorganisms have evolved in the oral cavity to colonize specific niches and build up the oral microbiome, one of the human body’s most complex and dynamic microbial communities [1]. The oral microbiota is composed of more than 700 species of microbes [2] whose association in consortia or competition dictates the health of the oral cavity, the progression from dental plaque to periodontitis [3], or even systemic chronic diseases, including cardiovascular diseases, diabetes, rheumatoid arthritis, cancers, and Alzheimer’s disease [4]. Potential pathogens such as *Streptococcus mutans* are commonly identified in the saliva and are considered initially colonizing in dental plaque [5]. In periodontal disease, culture-dependent and independent approaches have identified polymicrobial biofilm inhabited by the pathogenic triad (*Porphyromonas gingivalis*, *Tannerella forsythia*, *Treponema denticola*), Gram-positive bacteria (namely, *Filifactor alocis* and *Peptoanaerobacter stomatis*), Gram-negative bacteria of the Firmicutes phylum, and species of the genera *Prevotella*, *Desulfobulbus*, and *Synergistes*. Pathogenic bacterial consortia synergize to destroy the host response and dysregulate the composition of the healthy oral microbiota [6], composed of the genera *Streptococcus* (i.e., *S. mitis*, *S. oralis*, *S. sanguinis*, and *S. pneumoniae*), *Gemella*, *Rothia*, and *Prevotella*. 

*S. mutans* is deemed a key pathogen in the initiation and progression of caries because of its adhesion ability, the robust biofilm it forms on tooth surfaces, the rapid metabolism of a wide variety of carbohydrates, and the ability to resist the frequent environmental challenges encountered in oral microbial communities [7]. However, metagenome and metatranscriptome analyses have revealed that *S. mutans* accounts for less than 1% of the microorganisms in caries lesions [8], thus revolutionizing the understanding of the pathogenesis of oral diseases. It is well-accepted that even a few bacterial cells can modify the interaction patterns within a microbial community, producing virulence factors and leading to complex downstream effects on the host [9]. Recently, *S. mutans* has been considered one of the best-studied examples of inter- and intra-species bacterial communication, meaning the ability of bacteria to produce soluble factors, namely, the autoinducers, to interact with cells of the same strain or different bacterial species [10]. In bacteria, the production of autoinducers is mediated by Quorum Sensing, a cell-to-cell communication system that modifies the expression of several genes based on microbial population density and, consequently, coordinates the behavior of the bacterial population such as asbacteriocin production and biofilm formation. In particular, biofilms are clusters of bacteria of the same or different species attached to a surface and embedded in a self-produced matrix that protects bacteria from antibiotics or host immune cells [10]. *S. mutans* uses multiple Quorum Sensing (QS) systems to interact with other bacterial species in the oral cavity and to regulate biofilm formation, acid tolerance response, and genetic transformation [10]. The first QS system relies on the competence-stimulating peptide (CSP) encoded as a precursor by the loci *comCDE* [10]. The CSP precursor is cleaved to a 21-residue peptide further modified by an extracellular protease to remove the C-terminal 3 residues and generate the 18-residue functional peptide. When CSP reaches a critical concentration, it interacts with the ComD receptor protein of the neighboring bacterial cells and activates the response regulator, ComE, through autophosphorylation. The phosphorylated ComE activates downstream genes, triggering the signaling cascade for bacteriocin production, stress response, and biofilm formation [11,12,13]. A second QS system in *S. mutans* is activated by pathogens of the periodontal pockets. It relied on the sigX-inducing peptide (XIP) synthesized by ComS and sensed through ComR, an Rgg-type intracellular transcriptional regulator [14,15]. Rgg regulators perceive and respond to linear short hydrophobic peptides (SHPs) to regulate biofilm formation, capsule production, resistance to host factors, and competence [12]. 

Apparent homologs of the QS systems of *S. mutans* have been identified in the commensal Salivarius group of *Streptococcus* and *S. pneumoniae*, an opportunistic pathogen that participates in oral microbiota following aspiration [16,17]. Like in *S. mutans*, the QS system of *S. pneumoniae* controls competence and bactericidal activities by activating the ComCDE transductional pathway and CSP signal [12,18]. The production of bacteriocins is also regulated by the BlpRH pathway, which shares a high degree of sequence homology with *S. mutans* [19]. Upon CSP stimulation, the bacteriocin-like peptide (Blp) activates the histidine kinase receptor BlpH, which phosphorylates the response regulator BlpR. In *S. pneumoniae*, the Blp pathway results in the expression of genes involved in the Bacteriocin Immunity Region regulating antimicrobial sensitivity. 

Although the pheromone signals of the members of the genera *Streptococcus* are similar, their QS systems differ in the kinetic of virulence development, the aminoacidic sequences of the active CSP, the requirement of proteases for CSP activation, and the specificity for the intracellular sensors [9,20,21,22]. Therefore, CSP biosynthesis and maturation follow different pathways in *Streptococcus* spp. The impact on the dispersion, half-life, or interaction of QS signals generated from *Streptococcus* species at the intra- or inter-communication levels remains to be elucidated. Indeed, a better knowledge of the interaction of the CSP produced by a bacterial species and the receptor systems of a recipient bacterium may allow for interventions based on synthesized analogs of the peptide that would be of particular interest as postbiotic interventions in the regulation of virulence factors.

To better understand the interference of the QS systems in *Streptococcus* species, in this study, we investigated (i) the existence of CSP-mediated crosstalk between *S. mutans* and *S. pneumoniae*; (ii) the possibility of shutting down the virulence in *S. mutans* using altered sequences of the CSP produced by *S. pneumoniae*. Previous studies on the structure−activity relationships reported the possibility of using the altered CSP of *S. mutans* to modulate its QS [23,24]. However, no one has explored the option of exploiting the overlapping QS systems of intra-species communication as a new approach in quorum quenching. Here, we designed and synthesized analogs of *S. pneumoniae* CSP. Using in vitro assays, we demonstrated that the synthetic CSPs of *S. pneumoniae* inhibit bacteriocin production and biofilm formation in *S. mutans*. Our findings identified specific aminoacidic residues of the CSP sequence governing the conflict between these *Streptococcus* species, opening new implications in the biotechnological and biomedical fields. 

## 2. Materials and Methods

### 2.1. Chemicals 

The Fmoc-Lys(Boc) Wang resin, Acetonitrile (CH_3_CN), acetic anhydride, N, N-dimethylformamide (DMF), diethyl ether, 2,6-Lutidine, and triethylsilane (TES) were purchased from Sigma-Aldrich (Steinheim, Germany). N-hydroxybentrizol (HOBt) and 3-hydroxytriazolo[4,5-b]pyridine (HOAt) were purchased from Advanced Biotech Italia (Milan, Italy). All the Fmoc protected amino acids (AA), (E)-(((1-Cyano-2-ethoxy-2-oxyethylene)amino)oxy)tri(pyrrolidine-1-yl)phosphonium hexafluorophosphate(V) (PyOxim), 1-[Bis(dimethylamino)methylene]-1H-1,2,3-triazolo[4,5-b]pyridinium 3-oxid hexafluorophosphate (HATU), and N,N,N′,N′-Tetramethyl-O-(1H-benzotriazole-1-yl)uronium hexafluorophosphate (HBTU), were purchased from Novabiochem (Merck KGaA, Darmstadt, Germany). Trifluoroacetic acid (TFA), N-methyl-2-pyrrolidone (NMP), piperidine, N, N-Diisopropylethylamine (DIPEA), and Dichloromethane were purchased from Biosolve Chimie (Valkenswaard, The Netherlands).

### 2.2. Synthesis of the Peptide CSP1

The natural CSP peptide of *S. pneumoniae* CSP1 (sequence: H-Glu-Met-Arg-Leu-Ser-Lys-Phe-Phe-Arg-Phe-Ile-Leu-Gln-Arg-Lys-Lys-OH; see Table 1) was synthesized on Fmoc-Lys(Boc) Wang resin (0.44 mmol/g) using Fmoc chemistry by a Syro I synthesizer (Multisyntech, Witten, Germany). The side-chain protecting groups were OtBu, Boc, Trt, Pbf, and tBu. The coupling condition of each step is reported in Table 2. After the Fmoc-deprotection of the last inserted amino acid, the peptide was cleaved from the resin and deprotected from the side-chain protecting groups using the following mixture: 4.75 mL TFA, 0.125 mL TES, and 0.125 H_2_O for 1.5 h at room temperature. The resin was filtered off, and the solution was concentrated and added with cold diethyl ether. The product was precipitated and filtered. The identity of the peptide was determined by mass spectrometry (theoretical mass = 2242.69; experimental mass = 2242.58; SCIEX TOF-TOF 4800 instrument, International Equipment Trading Ltd., Mundelein, IL, USA). The peptide CSP1 was purified by RP-HPLC (purity grade > 95%) and characterized by analytical RP-HPLC (condition: Vydac C18 column (5 μm, 300 Å, 4.6 × 205 mm, Grace); eluent A: 0.05% TFA in H_2_O; eluent B: 0.05% TFA in CH_3_CN; gradient: from 25 to 45% B in 20 min; flow rate: 1 mL/min; detector at 214 nm; t_R_ = 11.268 min).

### 2.3. Synthesis of the Peptide AcCSP1

The peptide AcCSP1 (sequence: Ac-Glu-Met-Arg-Leu-Ser-Lys-Phe-Phe-Arg-Asp-Phe-Ile-Leu-Gln-Arg-Lys-Lys-OH; Table 1) was synthesized on Fmoc-Lys(Boc) Wang resin (0.44 mmol/g) using Fmoc chemistry by a Syro I synthesizer (Multisyntech; Witten, Germany). The side-chain protecting groups were OtBu, Boc, Trt, Pbf, and tBu. The coupling details are reported in Table 2. After Fmoc-deprotection of the last aminoacid, the peptide was acetylated on the resin by adding 10% *v*/*v* acetic anhydride and 5% lutidine in DMF for 20 min. After the acetylation reaction, the peptide was cleaved from the resin and deprotected from the side-chain protecting groups using the mixture of 4.75 mL TFA, 0.125 mL TES, and 0.125 mL H_2_O for 1.5 h at room temperature. The resin was filtered off and the solution was concentrated. The product was precipitated by the addition of cold ethyl ether and filtered. The identity of the peptide was determined by mass spectrometry (experimental mass = 2285.73; theoretical mass = 2288.36; SCIEX TOF-TOF 4800 instrument, International Equipment Trading Ltd., Mundelein, IL, USA). The peptide AcCSP1 was isolated by RP-HPLC (purity grade > 95%) and characterized by analytical RP-HPLC (condition: Vydac C18 column (5 μm, 300 Å, 4.6 × 205mm, Grace); eluent A: 0.05% TFA in H_2_O; eluent B: 0.05% TFA in CH_3_CN; gradient: from 25 to 35% B in 20 min; flow rate: 1 mL/min; detector: 214 nm; t_R_ = 8.54 min).

### 2.4. Synthesis of the PeptideAc4CSP1

The peptide Ac4CSP1 (sequence: Ac-Glu-Met-Arg-Leu-Ser-Lys(Ac)-Phe-Phe-Arg-Asp-Phe-Ile-Leu-Gln-Arg-Lys(Ac)-Lys(Ac)-OH; Table 1) was synthesized on Fmoc-Lys(Boc) Wang resin (0.44 mmol/g) using Fmoc chemistry by a Syro I synthesizer (Multisyntech; Witten, Germany). The side-chain protecting groups were OtBu, Boc, Trt, Pbf, and tBu. All the couplingsconditions are reported in Table 2. After the Fmoc-deprotection of the last amino acid, the peptide N-terminus was acetylated on the resin by adding 10% *v/v* acetic anhydride and 5% lutidine in DMF for 20 min. After the acetylation reaction, the peptide was cleaved from the resin, and the side-chain protecting groups were deblocked using the mixture of 4.75 mL TFA, 0.125 mL TES, and 0.125 mL H_2_O for 1.5 h. The resin was filtered off, and the solution was concentrated and added with cold diethyl ether. The product was precipitated and filtered. A second acetylation step was performed on the deprotected peptide under the same conditions indicated above. The identity of the peptide was determined by mass spectrometry (theoretical mass = 2410 Da; experimental mass = 2410.12 Da; SCIEX TOF-TOF 4800 instrument, International Equipment Trading Ltd., Mundelein, IL, USA). The peptide Ac4CSP1 was purified by RP-HPLC (purity > 95%) and characterized by analytical RP-HPLC (condition: Vydac C18 column (5 μm, 300 Å, 4.6 × 205 mm, Grace); eluent A: 0.05% TFA in H_2_O; eluent B: 0.05% TFA in CH_3_CN; gradient: from 30 to 45% B in 30 min; flow rate: 1 mL/min; detector: 214 nm; t_R_ = 15.496 min).

### 2.5. Synthesis of the Peptide CSP1hβE

The peptide CSP1hβE (sequence: H-hβGlu-Met-Arg-Leu-Ser-Lys-Phe-Phe-Arg-Asp-Phe-Ile-Leu-Gln-Arg-Lys-Lys-OH; Table 1) was synthesized on Fmoc-Lys(Boc) Wang resin (0.44 mmol/g) using Fmoc chemistry by a Syro I synthesizer (Multisyntech; Witten, Germany). The side-chain protecting groups were OtBu, Boc, Trt, Pbf, and tBu. The coupling details are reported for each step in Table 2. The cleavage from the resin was carried out, as indicated in previous paragraphs. The identity of the peptide was determined by mass spectrometry (theoretical mass = 2256 Da; experimental mass = 2258.07 Da; SCIEX TOF-TOF 4800 instrument). The peptide CSP1hβE was purified by RP-HPLC (purity grade > 95%) and characterized by analytical RP-HPLC (condition: Vydac C18 column (5 μm, 300 Å, 4.6 × 205 mm, Grace); eluent A: 0.05% TFA in H_2_O; eluent B: 0.05% TFA in CH_3_CN; gradient: from 25 to 35% B over 20 min; flow rate: 1 mL/min; detector: 214 nm; t_R_ = 9.483 min).

### 2.6. Synthesis of the Peptide CSP1AA

The peptide CSP1AA (sequence: H-Ala-Met-Ala-Leu-Ser-Lys-Phe-Phe-Arg-Asp-Phe-Ile-Leu-Gln-Arg-Lys-Lys-OH; see Table 1) was synthesized on Fmoc-Lys(Boc) Wang resin (0.44 mmol/g) using Fmoc chemistry by a Syro I synthesizer (Multisyntech; Witten, Germany). The side-chain protecting groups were OtBu, Boc, Trt, Pbf, and tBu. The coupling details are reported for each step in Table 2. The cleavage from the resin was carried out as indicated in previous paragraphs. The identity of the peptide was determined by mass spectrometry (theoretical mass = 2099 Da; experimental mass = 2099.58 Da; SCIEX TOF-TOF 4800 instrument, International Equipment Trading Ltd., Mundelein, IL, USA). The peptide CSP1AA was purified by RP-HPLC (purity > 95%) and characterized by analytical RP-HPLC (condition: Vydac C18 column (5 μm, 300 Å, 4.6 × 205 mm, Grace); eluent A: 0.05% TFA in H_2_O; eluent B: 0.05% TFA in CH_3_CN; gradient: from 28 to 38% B over 20 min; flow rate: 1 mL/min; detector: 214 nm; t_R_ = 7.535 min).

### 2.7. Synthesis of the Peptide CSP1Y(SO_3_)

The peptide CSP1Y(SO_3_) (sequence: H-Tyr(SO_3_H)-Met-Arg-Leu-Ser-Lys-Phe-Phe-Arg-Asp-Phe-Ile-Leu-Gln-Arg-Lys-Lys-OH; see Table 1) was synthesized on Fmoc-Lys(Boc) Wang resin (0.44 mmol/g) using Fmoc chemistry by a Syro I synthesizer (Multisyntech; Witten, Germany). The side-chain protecting groups were OtBu, Boc, Trt, Pbf, and tBu. The coupling details are reported for each step in Table 2. The cleavage from the resin was carried out as indicated in previous paragraphs. The identity of the crude peptide was determined by mass spectrometry (theoretical mass = 2363.84 Da; experimental mass = 2357.3 Da; ESI-TOF, Mariner System 5220, Applied Biosystem, Perkin-Elmer, Foser City, CA, USA). The peptide CSP1Y(SO_3_) was isolated by RP-HPLC (purity > 95%) and characterized by analytical RP-HPLC (condition: Vydac C18 column (5 μm, 300 Å, 4.6 × 205 mm, Grace); eluent A: 0.05% TFA in H_2_O; eluent B: 0.05% TFA in CH_3_CN; gradient: from 25 to 35% B over 20 min; flow rate: 1 mL/min; detector: 214 nm; t_R_ = 11.827 min).

### 2.8. Bacterial Strains and Growth Conditions

*Streptococcus mutans* Clarke (strain designation NCTC 10449), *Pseudomonas aeruginosa* (Schroeter) Migula (strain designation PAO1), *Fusobacterium nucleatum*, subspecies *nucleatum* (strain VPI 4355), and *Porphyromonas gingivalis* (strain 2561) were purchased from ATCC (LGC Standards Srl, Milan, Italy). *Streptococcus mitis, Streptococcus oralis*, and *Streptococcus sanguinis* were obtained from clinical isolates and characterized by morphological, biochemical, and molecular typing. *S. mutans*, *S. mitis*, *S. oralis*, and *S. sanguinis* were maintained in Brain Heart Infusion (BHI) broth or agar under microaerophilic conditions. *P. aeruginosa* was maintained in trypticase soy (TS) agar or broth. *F. nucleatum* was maintained in TS agar or broth with defibrinated sheep blood under anaerobic conditions. *P. gingivalis* was maintained in ATCC Medium 2722 supplemented with TS broth or agar under anaerobic conditions. Before the experiments, bacterial cultures were inoculated in fresh media (dilution 1:100) and grown for 16 h at 37 °C.

### 2.9. Susceptibility Test

The antimicrobial activities of the tested peptides were assessed through the determination of the minimum inhibitory concentration (MIC) standardized by the Clinical and Laboratory Standards Institute [25,26]. Bacterial inocula were diluted in 10 mL of MHB (medium composed of milk, honey, and bromothymol blue) [27]. The concentration of the bacteria was adjusted to 0.5 McFarland standard, and the bacteria were dispensed in 96-well microtiter plates (final bacterial concentration: 1 × 10^4^ CFU/well). Peptides were dissolved in sterile water (stock concentration: 2 mM) and added at final concentrations ranging from 0 to 50 µM. Plates were incubated at 37 °C for 16 h. At the end of incubation, bacterial growth was recorded using a microplate reader (Varioskan Lux Reader; Thermo Fisher Scientific, Milan, Italy) by quantifying the adsorption at 620 nm. Cultures were then opportunely diluted and seeded on agar plates. Bacterial cultures inoculated with the highest final concentration of the vehicle (sterile water) or chlorhexidine hydrochloride salt (CHX, concentration range: 0.05 to 500 μg/mL; Merck, Milan, Italy) were included as controls in each test. Experiments were performed in triplicate.

### 2.10. Bacterial Growth Kinetics

Peptides were further evaluated for their potential effect in altering the growth kinetics in *S. mutans*. Overnight bacterial cultures were collected, centrifuged, and suspended in fresh BHI broth at 10^8^ CFU/mL. Bacteria were dispensed in sterile 96-well microtiter plates at a final concentration of 1 × 10^6^ CFU/100 µL and cultured with peptides or the vehicle at 37 °C for 36 h. Bacterial growth kinetics were monitored by measuring the optical density at 620 nm at different time points using Varioskan Lux Reader (Thermo Fisher Scientific).

### 2.11. Cultures of Human Gingival Fibroblasts and Cytotoxicity Assay

Human gingival fibroblasts (HGFs) were prepared as previously described [28]. Cells were obtained from gingiva samples collected during a surgical procedure from a young adult healthy woman with her explicit informed consent (research protocol approved by the ethical committee of the Azienda Ospedaliera/Università di Padova Aut. 4899/AO/20). Gingiva samples were cut into small pieces, and cells were isolated by digestion with collagenase and trypsin. HGF were cultured in Dulbecco’s Modified Eagle Medium (DMEM) supplemented with 10% *v/v* fetal bovine serum, 2 mM L-glutamine, and 1% *v/v* penicillin-streptomycin (all purchased from Thermo Fischer Scientific, Milan, Italy). The purity of the cell cultures was ascertained by fluorescence-activated cell sorting (FACS) analysis using anti-vimentin antibody (clone VM452; Genetex International provided by Prodotti Gianni, Milan, Italy) and by confocal microscopy using anti-actin α-smooth muscle antibody (clone 1A4; Merck; Milan, Italy). The purity of cell cultures was reported as greater than 97% (data not shown).

At the confluence, HGFs were detached and seeded on 96-well plates (1 × 10^3^ cells/well). Twenty-four hours later, cells were added with peptides and incubated for an additional 24 h. To exclude the cytotoxic effects of peptides on HGFs, cells were incubated with 0.5 mg/mL 3-[4,5-dimethylthiazole-2-yl]-2,5-diphenyltetrazolium bromide (MTT; Merck) solution for 4 h at 37 °C. Formazan crystals were solubilized in 100 µL of SDS 10% *w/v* and HCl 0.01 N. The absorbance was recorded 16 h later at 590 nm using a microplate reader (Varioskan Lux Reader; Thermo Fisher Scientific).

### 2.12. Assessment of Anti-Biofilm Activities

Overnight cultures of *S. mutans* were diluted 1:1000 in a 1:10 solution of BHI broth supplemented with 0.2% *w/v* sucrose and artificial saliva (AS; 1.3 g/L KCl, 0.1 g/L NaCl, 0.05 g/L MgCl, 0.1 g/L CaCl_2_, 0.5 g/L peptone, 25 μg/L NaF, 27 mg/L KH_2_PO_4_, 35 mg/L K_2_HPO_4_, pH 6.8; all reagents purchased from Merck) [29]. Cultures were seeded into 96-well polystyrene microtiter plates (200 µL/well) and incubated at 37 °C under static conditions. To identify the time of incubation that is useful in obtaining a mature biofilm, plates were incubated for 16, 24, 48, 72, or 96 h. Half of the media was renewed every 24 h. At the end of incubations, the wells were emptied and washed three times with sterile PBS to remove planktonic cells. Adhering cells were incubated with resazurin 0.01% in the dark at 37 °C for 20 min [30]. In metabolically active cells, the non-fluorescent resazurin was reduced to highly fluorescent resorufin by dehydrogenase enzymes. The relative fluorescence units (RFU) of resorufin were measured using a fluorimeter (Ex = 530–570 nm, Em = 590–620 nm; Varioskan Lux Reader; Thermo Fisher Scientific).

To assess the effect of the compounds on biofilm formation or mature biofilm, cultures of *S. mutans* were prepared as described above. The peptides were added at different time points, and cultures were incubated at 37 °C. At the end of incubation, the culture media were discarded, and the wells were washed three times with sterile PBS to remove planktonic cells. Adhering bacteria were stained with 150 µL of 0.1% (*w/v*) crystal violet (CV; Merck, Milan, Italy) solution for 15 min at room temperature. The plates were washed and air dried, and the CV was solubilized in 125 µL of 30% *v/v* glacial acetic acid. The optical density was measured at 570 nm using a microplate reader (Varioskan Lux Reader; Thermo Fisher Scientific). We assigned 100% biofilm formation to bacteria incubated with the vehicle alone. All the experiments were performed three times with triplicate determinations for each condition.

### 2.13. Bacteriocin Assays

During the formation of complex biofilm such as dental plaque, the ability of *S. mutans* to compete with other commensal species determines the success of the cariogenesis process. *S. mutans* uses antibacterial peptides, namely, bacteriocins, to overcome and subvert the composition of dental microbiota [31,32]. We investigated the production of bacteriocins using the overlay assay [33]. Overnight cultures of *S. mitis*, *S. oralis*, *S. sanguinis*, *P. aeruginosa*, *F. nucleatum*, or *P. gingivalis* were diluted 1:100 in BHI or TS broth and incubated at 37 °C until they reached an OD_620_ of 0.3. Cultures were added with agar (BD Difco^TM^ Agar) at the final concentration of 1.5% *w/v* and transferred in plates. Overnight cultures of *S. mutans* were diluted 1:100 in BHI and added with peptides (20 μM) or the vehicle with or without CSP1. Culture media were supplemented with agar to reach the final concentration of 0.7% *w/v*. The solutions were spotted on the surface of agar plates prepared as described above. The plates were kept on the benchtop for 30 min, allowing the agar containing *S. mutans* to solidify, and then incubated at 37 °C for 16 h under microaerophilic or anaerobic conditions, as required.

Spots (10 for each experimental group) were acquired and analyzed using NIH Image J software (version 1.53t August 2022, https://imagej.nih.gov/ij/index.html; accessed on 7 September 2022). The appearance of a clear area on the plate (bacterial growth inhibition) was evaluated as indicative of bacteriocin activity. The diameters of the area were measured in a blinded fashion.

### 2.14. Total RNA Isolation and Quantitative RT-PCR

*S. mutans* was cultured in BHI supplemented with peptides or the vehicle. Cultures were centrifuged, and total RNA was isolated and purified from the bacterial pellets using the GRS Total RNA Kit—Bacteria (GK16.0100; GRISP Research Solution, Porto, Portugal), following the manufacturer’s instructions. Purified RNA was subjected to DNase I treatment to remove contaminating DNA. The RNA yield and purity were assessed by measuring the absorbance, and only samples with a ratio of 260/280 nm in the 1.8–2 range were used. cDNA was generated using the cDNA RT kit with an RNAse inhibitor (Applied Biosystems; Monza, Milan). Quantitative PCR (qRT-PCR) was performed using SYBR green mixture (iScript One Step RT-PCR kit; Bio-Rad, Segrate, Milan) to determine the transcript levels of genes using the oligonucleotides listed in Table 3. The *gyrA* gene served as the housekeeping gene.

### 2.15. Data Analysis

The results are reported as the mean ± SEM. Statistical tests were performed using GraphPad Prism v.9.4.0 software. Analyses of variances (ANOVA) were performed, followed by Bonferroni’s post hoc test or a two-tailed paired or Student’s *t*-test, as appropriate. Values of *p* < 0.05 were regarded as significant.

## 3. Results

### 3.1. Minimum Inhibitory Concentrations of Peptides

As determined by the microdilution method, the peptides reported MIC values ranging between 20 and 30 μM when tested against *Streptococcus* strains (Table 4). Concentrations of peptides higher than 40 μM reported bactericidal effects, as determined by the colony counting assay. Conversely, MIC values were greater than 50 μM when the peptides were tested against *P. aeruginosa*, *F. nucleatum*, and *P. gingivalis* (Table 1). As a positive control, CHX reported higher MIC values in *F. nucleatum* (300 μg/mL), whereas induced bactericidal activities at concentrations below 1 μg/mL in *S. mutans*, *S. mitis*, *S. oralis*, and *S. sanguinis* were determined by the microdilution method (Table 4) and colony counting assay. Considering the MIC results, we used peptides at a concentration of 20 μM in the experiments described below.

### 3.2. Peptides Do Not Interfere with the Viability of S. mutans and Human Fibroblasts

To exclude the direct toxic effects, we incubated *S. mutans* with the natural peptide CSP1 (20 µM) of *S. pneumoniae*. The optical density of the bacterial growth was monitored for 36 h to measure the microbial growth curve. As reported in Figure 1a, CSP1 of *S. pneumoniae* at the tested concentration had no significant effects on the growth of *S. mutans*. At the same time, the colony counting assay was not affected by incubation with the peptide (data not shown). Similarly, CSP1 analogs did not alter the kinetic growth of *S. mutans* (Figure 1a).

Thinking about a possible therapeutic application in oral mucosa, we investigated the effects of the peptides on the viability of human primary fibroblasts. Thus, eukaryotic cells were exposed to 20 μM peptides for 24 h. Cell viability was determined by the MTT test. As reported in Figure 1b, the peptides did not report cytotoxic activity. Overall, the peptides showed no direct toxicity on prokaryotic and eukaryotic cells.

### 3.3. Peptides Reduce Biofilm Formation in S. mutans

*S. mutans* is considered a major contributor during cariogenesis because of its ability to produce biofilm [34]. Our experiments reported that, in cultures of *S. mutans* grown in 1:10 BHI:AS media, the number of adherent bacterial cells increased following 48 h of incubation, peaked at 72 h, and then reduced at 96 h (Figure 2a). To evaluate the interference of *S. pneumoniae* QS regarding *S. mutans* biofilm formation, bacteria were added with 20 μM peptides and incubated for 72 h. Media and stimuli were renewed every 24 h. Surprisingly, we observed that natural CSP1 significantly increased the ability of *S. mutans* in producing biofilm, as compared with cultures incubated with the vehicle (*p* < 0.05; Figure 2b). The synthesized peptides reduced biofilm formation as compared with cultures stimulated with CSP1. In addition, CSP1AA reduced biofilm formation by 44% in comparison with *S. mutans* incubated with the vehicle alone (*p* < 0.05; Figure 2b). As the positive control, CHX reduced biofilm by 70% vs. cultures incubated with the vehicle.

Finally, to assess the role of peptides on mature biofilm, *S. mutans* was cultured as described, and peptides were added 48 h later. Biofilm was assessed at 72 h of culture. The peptides did not affect the mature biofilm as compared with the vehicle, whereas CHX reduced the biofilm by 54% (Figure 2c).

### 3.4. Peptides AcCSP1, Ac4CSP1, and CSP1AA Counteract the Bacteriocin Production in S. mutans

We initially performed the overlay assays by incubating *S. mutans* with 20 μM peptides, as described in Materials and Methods. We observed that the stimulation of *S. mutans* with the natural CSP1 of *S. pneumoniae* significantly inhibited the growth of the co-cultured *Streptococcus* spp. (Figure 3a), as denoted by the enlarged diameter of growth inhibition. These data indicate that *S. pneumoniae* mediates interference in the QS of *S. mutans* affecting the production of bacteriocins. Cultures of *S. mutans* incubated with the vehicle did not report inhibitory effects on bacteriocin production (Figure 3a). Therefore, we decided to force the production of bacteriocins in *S. mutans* by incubating cultures of *S. mutans* with the natural CSP1 of *S. pneumoniae* to test the effects of the CSP1 analogs (each one at 20 μM) and evaluate their inhibitory effects on bacteriocin production.

As reported in Figure 3b–d, the synthesized AcCSP1, Ac4CSP1, and CSP1AA peptides significantly reduced the antibacterial effects induced by CSP1 against commensal *S. mitis*, *S. oralis*, and *S. sanguinis*. Competition among the natural CSP1 and CSP1 analogs could explain the results. However, the peptides CSP1hβE and CSP1Y(SO_3_) did not report significant effects. *F. nucleatum* and *P. gingivalis* are pathogenic bacteria that participate in the progression of dental disease and the formation of periodontitis. We revealed that the natural peptide CSP1 inhibited the growth of *F. nucleatum* (Figure 3e) but had no effects on *P. gingivalis*. The synthesized peptides did not affect the CSP1-induced inhibition of *F. nucleatum* (Figure 3e). *P. aeruginosa* was used as an experimental control. It grew well under microaerophilic conditions, and, as expected, the replication of the bacterium was not affected by the presence of *S. mutans* stimulated by the native CSP1.

### 3.5. The Synthesized Peptides Inhibit the Expression of Virulence Genes

We investigated the impact of the synthesized peptides of *S. pneumoniae* on the transcription of genes involved in the virulence of *S. mutans*, such as bacteriocin production and biofilm formation. At first, *S. mutans* were cultured with the vehicle or natural CSP1 for 2–24 h, and the expression of *bip*, a gene involved in antimicrobial activity and the export of bacteriocins [35], was determined by qPCR. As reported in Figure 4a, the natural peptide CSP1 of *S. pneumoniae* induced the expression of *bip*, and maximum levels of gene expression were recorded following 8 h in culture. Therefore, we incubated *S. mutans* with CSP1 and the CSP1 analogs for 8 h. We revealed that AcCSP1, Ac4CSP1, and CSP1AA significantly reduced the CSP1-induced expression of *bip* (Figure 4b).

The VicRKX system of *S. mutans* plays an important role in biofilm formation. In particular, the *vicK* gene is involved in the production of proteins of the extracellular matrix [36]. In the biofilm of *S. mutans* stimulated with CSP1, the expression of the *vicK* gene significantly increased following 16 h of incubation (Figure 4c). All the synthesized peptides reported the tendency to reduce the CSP1-dependent *vicK* gene expression, but only the peptide CSP1AA reported a significant inhibition (*p* < 0.05 vs CSP1; Figure 4d).

To better investigate the involvement of the QS of *S. pneumoniae* in the antivirulence activity of the mutated peptides, we evaluated the expression of *comCDE* genes in cultures of *S. mutans*. As reported in Figure 4e, following 8 h of stimulation, the CSP1 of *S. pneumoniae* increased the expression of *comD* and *comE* genes but had no effects on the *comC* gene expression of *S. mutans*. Incubation with the peptides AcCSP1, Ac4CSP1, and CSP1AA inhibited the expression of *comD* and *comE* induced by CSP1.

## 4. Discussion

In multi-species microbial communities, bacteria engage the quorum sensing (QS) systems to coordinate behaviors at multicellular levels and regulate cooperation or competition, thus selecting a group of bacteria that benefits from the environmental traits [37]. At the intra-species levels, the clonal nature of bacteria in the community facilitates cooperative behaviors, whereas metabolic networks associate microbes of different genera [36]. However, competition for a limiting resource triggers kin selection in bacteria sharing common metabolic pathways, and social cheaters can emerge [38,39]. Metabolic competition and quorum sensing-derived signals are potent drivers in the diversification and dynamic composition of the microbial communities, as well as in the interaction with the host [1,10,40]. The mouth provides a carbohydrate-enriched environment ideal for colonization by different bacteria. Streptococci, in particular, are the dominant species producing QS signals in the human oral cavity [10,41]. The abundance and metabolic activity of Streptococci are related to common human diseases, namely, dental caries and periodontitis. Microbiome studies identified acid-producing and acid-tolerant bacterial species, including *S. mutans*, in dentin carious lesions, whereas *S. sanguinis*, *S. mitis*, *S. intermedius*, and *S. infantis* are associated with healthy sites [42]. Streptococci share common quorum sensing systems but evolved different metabolic pathways. Thus, two-component signal transduction pathways (ComCDE and SigX) and specific competence stimulating peptides (CSP) control the competence and production of bacteriocins in *S. mutans* and *S. pneumoniae* [16]. However, *S. mutans* preserves the tricarboxylic acid cycle enzymes that *S. pneumoniae* lost. Under limited resources, this metabolic trait favors *S. mutans* in prioritizing carbohydrates and expressing virulence to increase the production of bacteriocins and trigger fratricide competition [16,43]. To better dissect the interactions between the QS of *S. mutans* and *S. pneumoniae*, in this study, we exposed cultures of *S. mutans* to analogs of the CSP produced by *S. pneuomoniae* and evaluated the effects on the competition among bacterial species associated with the healthy or cariogenic statuses. Since inhibiting the QS pathways in *S. mutans* modulates the microbiome [9], our findings open the possibility of exploiting the intra-species fratricide signals in treating complex microbial dysbiosis, such as the ones occurring in cariogenesis.

In this study, we used different experimental approaches to investigate the inter-relationships of the QS systems in *S. mutans* and *S. pneumoniae* (see Figure 5 for an infographic summary of the main experimental assays and results). We reported that the wild-type sequence of *S. pneumoniae* CSP activates the QS in *S. mutans*, as evidenced by the induction of biofilm and bacteriocins (Figure 2 and Figure 3). The mutated CSP peptides of *S. pneumoniae* significantly reduced the expression of *ComD* and *ComE* genes in *S. mutans* (Figure 4), indicating the ability to interfere with the sensing and processing of QS signals, respectively. Indeed, CSP analogs of *S. pneumoniae* inhibited the cell killing of *S. mutans* against normal commensal bacteria of the oral mucosa, namely, *S. mitis*, *S. oralis*, and *S. sanguinis* (Figure 3), the formation of biofilm (Figure 2), and the virulence factors controlled by the QS [16,43]. In *S. mutans*, the membrane-associated receptor ComD is a histidine kinase protein that binds the natural CSP and triggers the signaling cascade to express virulence. The membrane domain of ComD forms three extracellular loops, of which two (loopC and loopB) participate in CSP recognition [11,44]. LoopB and loopC consist of 15 amino acid residues. It has been reported that the deletion of LDGT residues in loopB reduced CSP-mediated QS activation, such as bacteriocin production, whereas the residues QGIV do not participate in CSP effects. In loopC, the sequence NVIP is mandatory in response to CSP, and the four residues TLKF reduced the CSP activity by 50% [11]. In this study, we designed and synthesized five *S. pneumoniae* CSP analogs characterized by residual modifications at positions 1, 3, 6, 16, and 17. Yang Y. et al. [45] demonstrated that Glu^1^ residue is fundamental because it drives receptor activation: the modification Glu^1^→D-Glu provides an analog without bioactivity, while the Glu^1^→Ala modification produces a ComD1 inhibitor. Our analogs, CSP1βhE (Glu^1^→β-homo-Glu) and CSP1Y(SO3) (Glu^1^→Tyr(SO3)), maintain the negative charge in the side chain modifying the structure of the amino acid, while in AcCSP1, the acetylation eliminates the positive charge of the original N-terminal amino group. In addition to the Glu^1^, another residue of fundamental importance in initial receptor recognition is the Arg^3^ residue. The analog CSP1AA (Glu^1^→Ala and Arg^3^→Ala) presents a double modification that turns off both key residues. To investigate the role of the C-terminus residues and confirm the scarce importance of the central hydrophilic region [45], we synthesized analogs bearing the acetylation of all the Lysines and the amino terminus (Table 1).

Several *S. mutans* strains have one or more point mutations in the *ComD* sequence coding extracellular loops, but they are equally able to sense CSP and activate the QS system [44,46,47]. These studies suggest that, in *S. mutans*, the CSP-ComD interaction is characterized by a relatively low specificity, allowing *S. mutans* to eavesdrop on the communication between Streptococci and eventually take the opportunity to outnumber them. The lower constraint specificity of ComD has been exploited in our study. Indeed, it is well recognized that QS signal molecules induce intra-species and inter-species effects, even at nanomolar concentrations [48,49]. Dong G. and colleagues reported that 0.5 μM natural CSP peptide activated *S. mutans* QS following 90 min of exposure [11]. In our experiments, we exposed *S. mutans* to 20 μM of “non-autologous” CSP (generated by *S. pneumoniae*) and its analogs, a concentration reporting the minimal inhibitory effects following 16 h of incubation in all the tested *Streptococcus* spp. (Table 4). Thus, the main goal of QS inhibition is to reduce the virulence of bacteria with no toxic effects (Figure 1) and no possibility of the occurrence of resistance [50]. We are running experiments to assess the anti-QS concentrations of CSP analogs better. However, the active concentration of autoinducers is challenging, as dilution effects and proteases produced by bacteria or the host make the QS signals fluctuate, even in narrow environments [11,34,47]. Moreover, in microbial communities, signal propagation does not follow linear diffusion, and short-range or temporal-limited increased concentrations of QS signals are enough to induce virulence [51].

Even if our study requires further experiments to investigate the effects of CSP analogs on the QS of commensal *Streptococcus* spp., our preliminary results report no interference of CSP analogs in the QS system of *S. pneumoniae*. Contrary to *S. mutans*, two major CSP variants activate the QS in *S. pneumoniae* by specific interaction with their receptors ComD1 and ComD2 [52]. Mutations in the CSP sequence abort the activation of the QS system in *S. pneumoniae*, thus excluding the ability of our CSP analogs to interfere with the virulence of the bacterium. In silico analysis suggest that ComD proteins of *S. pneumoniae* and *S. mutans* differ in the extracellular domains [52]. Moreover, phylogenetic analysis reveals that the ComCDE system of *S. mutans* combines the signals of two QS pathways of *S. pneumoniae* (ComCDE and BlpCRH, both involved in bacteriocin production), thus providing further evidence of the inter-relationships between the two Streptococcal species [53].

In conclusion, the higher flexibility of the ComD receptor in *S. mutans* allows it to sense signals produced by different kin bacterial species to provide advantages in the dental biofilm. Our findings improve the knowledge about the chemical moiety required by soluble microbial factors involved in the interaction between *S. mutans* and *S. pneumoniae*. Moreover, we provided evidence that it is possible to exploit the ComD receptor of *S. mutans* to restore intra-species competition in microbial communities, such as the oral cavity microbiota.

## Figures and Tables

**Figure 1 microorganisms-10-02386-f001:**
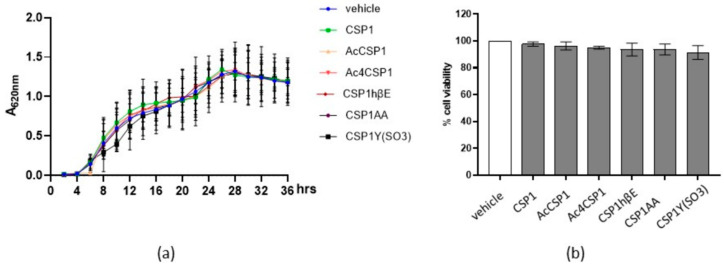
Effects of peptides on prokaryotic and eukaryotic cell growth and viability. (**a**) Cultures of *S. mutans* (1 × 10^6^ CFU/mL) were incubated with 20 µM of peptides or the vehicle. Bacterial growth was monitored for 36 h by measuring the absorbance at 620 nm (A620 nm). Experiments were performed three times. Data are reported as the mean ± SEM. (**b**) Human primary gingival fibroblasts (HGFs) were cultured with 20 μM peptides for 24 h. Cell viability was assessed by an MTT assay. Experiments were performed two times. Data are reported as the percentage of cell viability calculated over the control (vehicle).

**Figure 2 microorganisms-10-02386-f002:**
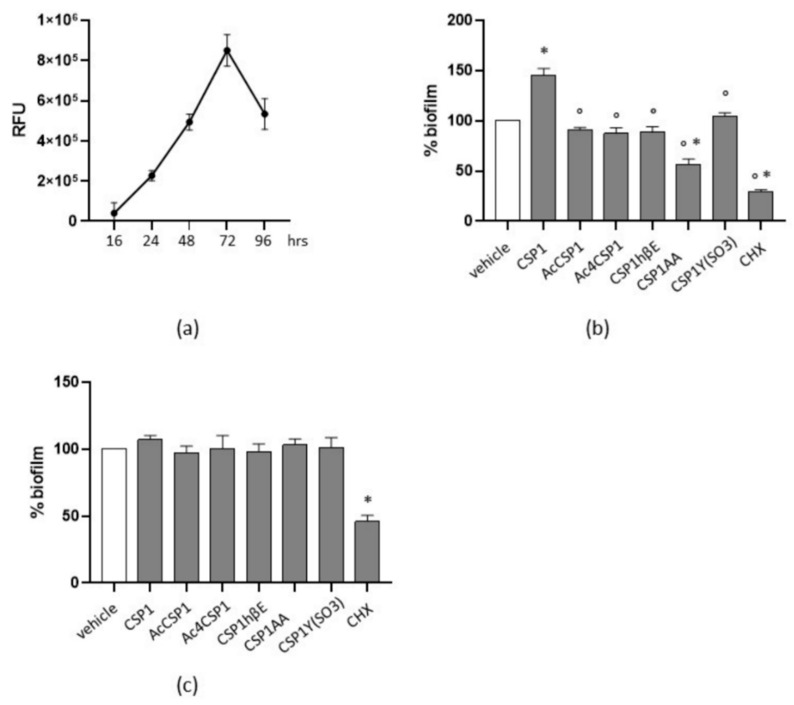
Peptides interfere in *S. mutans* biofilm formation. (**a**) Cultures of *S. mutans* were seeded in 96-well plates and incubated in 1:10 BHI:AS medium at 37 °C. Adhering cells were determined at the specified time points after incubation for 20 min with resazurin 0.01%. Biofilm formation was evaluated by measuring the relative fluorescence units (RFU) using a fluorimeter (Ex = 530–570 nm, Em = 590–620 nm). Experiments were performed three times. Data are reported as the mean ± SEM. (**b**) Percentage of residual biofilm. *S. mutans* was incubated with peptides (20 μM) or chlorhexidine (CHX, 0.63 μg/mL) for 72 h. Stimuli were renewed every 24 h. Biofilms were evaluated by crystal violet staining. Cultures incubated with the vehicle were assigned as 100% biofilm. Data are reported as the mean ± SEM of three independent experiments. (**c**) To evaluate the effects on mature biofilm, *S. mutans* was incubated for 48 h. CHX (0.63 μg/mL) and peptides (20 μM) were added, and biofilms were evaluated 24 h later by crystal violet staining. Cultures incubated with the vehicle were assigned as 100% biofilm. Data are reported as the mean ± SEM of three independent experiments. * denotes *p* < 0.05 vs. the vehicle; ° denotes *p* < 0.05 vs. CSP1.

**Figure 3 microorganisms-10-02386-f003:**
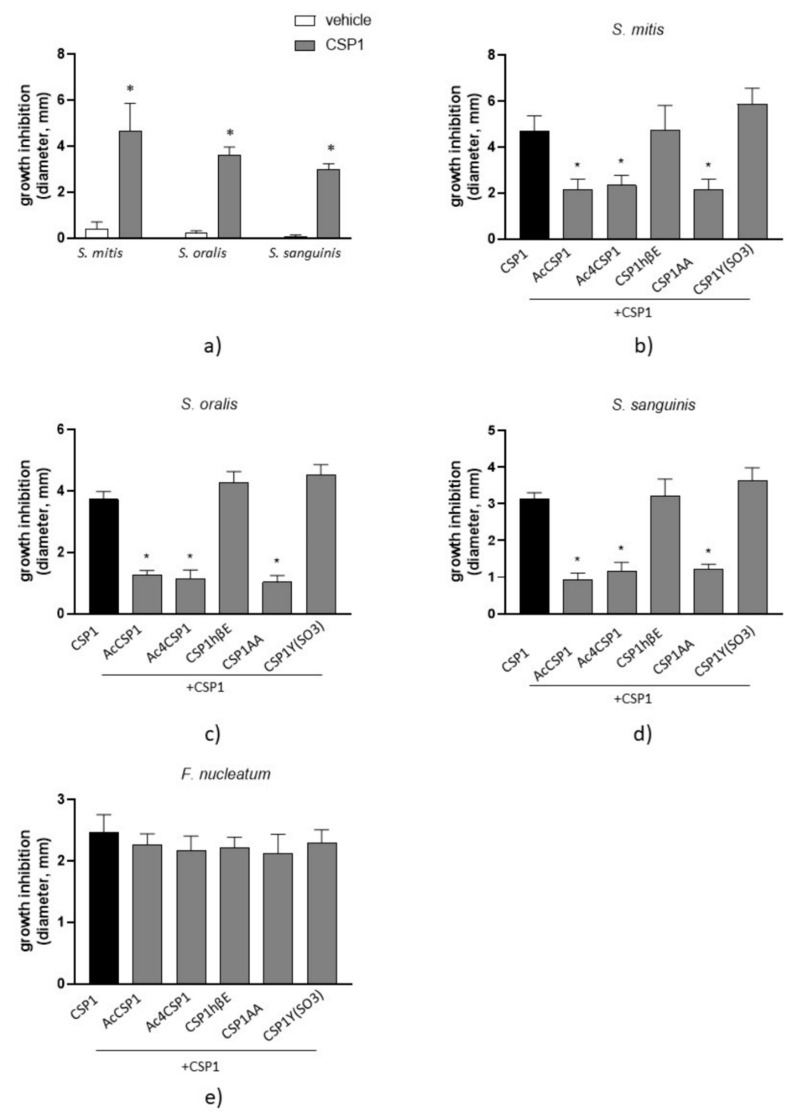
Bactericidal activity of synthesized peptides. (**a**) Cultures of *S. mutans* were incubated with the vehicle or CSP1 (20 μM) and spotted on 0.7% agar media on the surface of agar plates previously inoculated with *S. mitis*, *S. oralis*, or *S. sanguinis*. Plates were incubated under microaerophilic conditions at 37 °C for 16 h. Cultures of *S. mutans* were incubated with CSP1 and analogs (20 µM) and plated as described above. Cultures were spotted on agar plates inoculated with *S. mitis* (**b**), *S. oralis* (**c**), or *S. sanguinis* (**d**). Plates were incubated under microaerophilic conditions at 37 °C for 16 h. *S. mutans* cultures were spotted on agar plates inoculated with *F. nucleatum* and incubated under anaerobic conditions at 37 °C for 16 h (**e**). At the end of the incubations, the diameters of clear areas indicating bactericidal effects were measured. Data are reported as the mean ± SEM of at least five independent experiments, counting ten spots for each experimental group. * denotes *p* < 0.05 vs. CSP1.

**Figure 4 microorganisms-10-02386-f004:**
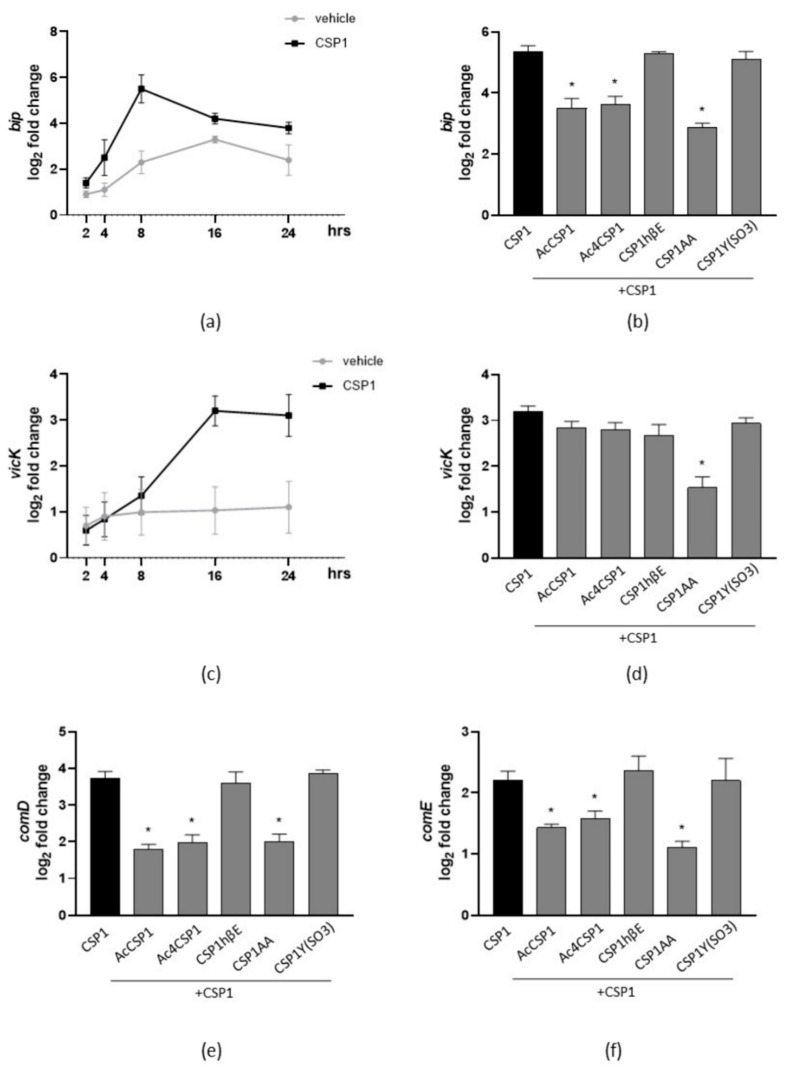
Effects of synthesized peptides on QS-related gene expression. (**a**) Cultures of *S. mutans* were incubated with the vehicle or CSP1 (20 μM) for 2–24 h. Bacteria were collected and subjected to RNA extraction. Expression of the *bip* gene was investigated by qPCR. (**b**) Cultures of *S. mutans* were incubated with CSP1 with or without the synthesized peptides (each one at 20 μM). The expression of the *bip* gene was investigated 8 h later by qPCR. (**c**) *S. mutans* were diluted in BHI containing 0.2% *w/v* sucrose and AS (1:10) media to obtain biofilm. Cultures were added with the vehicle or CSP1 (20 μM) and incubated for 2–24 h. Adherent bacteria were collected and subjected to RNA extraction. Expression of the *vicK* gene was investigated by qPCR. (**d**) *S. mutans* was incubated with CSP1 with or without synthesized peptides (each one at 20 μM) and cultured to obtain biofilm. The expression of the *vicK* gene was investigated 16 h later by qPCR. Planktonic cultures of *S. mutans* were incubated for 8 h with CSP1 with or without synthesized peptides (each one at 20 μM). Bacteria were collected and subjected to RNA extraction. Expressions of *comD* (**e**) or *comE* (**f**) genes were investigated by qPCR. The results are reported as the log_2_ of relative gene expression. The data are expressed as the mean ± SEM of three independent experiments. * denotes *p* < 0.05 vs. CSP1.

**Figure 5 microorganisms-10-02386-f005:**
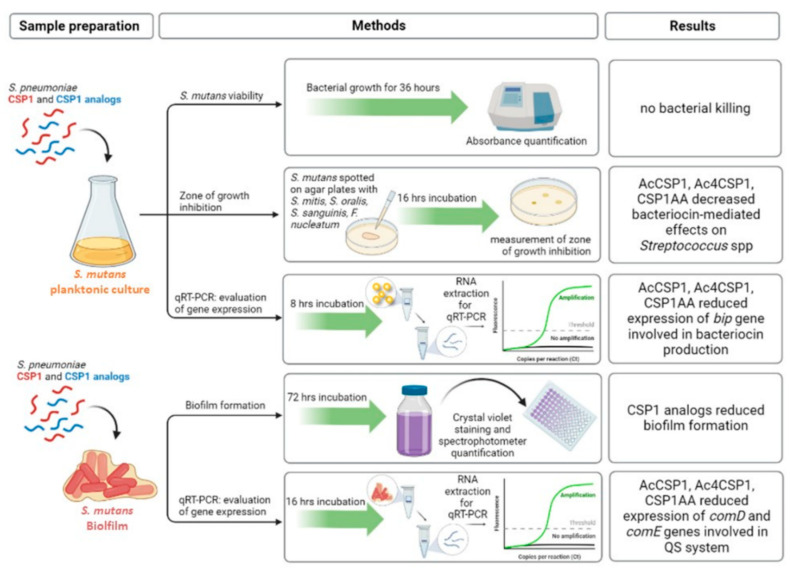
Schematic representation of the main experimental approaches and results obtained in the study. As indicated, *S. mutans* in planktonic or biofilm cultures was incubated with the natural CSP1 of *S. pneumoniae* or CSP1 analogs at different times. Cultures were then subjected to the measurement of cell viability by spectrophotometry, bacteriocin production by co-culture experiments, and the measurement of the zone of inhibition, biofilm evaluation, and qPCR to evaluate the expression of genes involved in the QS system.

**Table 1 microorganisms-10-02386-t001:** Sequences of the synthetic peptides.

Peptide Name	Peptide Sequence
CSP1	H-Glu-Met-Arg-Leu-Ser-Lys-Phe-Phe-Arg-Asp-Phe-Ile-Leu-Gln-Arg-Lys-Lys-OH
AcCSP1	**Ac**-Glu-Met-Arg-Leu-Ser-Lys-Phe-Phe-Arg-Asp-Phe-Ile-Leu-Gln-Arg-Lys-Lys-OH
Ac4CSP1	**Ac**-Glu-Met-Arg-Leu-Ser-Lys(**Ac**)-Phe-Phe-Arg-Asp-Phe-Ile-Leu-Gln-Arg-Lys(**Ac**)-Lys(**Ac**)-OH
CSP1hβE	H-**hβGlu**-Met-Arg-Leu-Ser-Lys-Phe-Phe-Arg-Asp-Phe-Ile-Leu-Gln-Arg-Lys-Lys-OH
CSP1AA	H-**Ala**-Met-**Ala**-Leu-Ser-Lys-Phe-Phe-Arg-Asp-Phe-Ile-Leu-Gln-Arg-Lys-Lys-OH
CSP1Y(SO_3_)	H-**Tyr(SO_3_H)**-Met-Arg-Leu-Ser-Lys-Phe-Phe-Arg-Asp-Phe-Ile-Leu-Gln-Arg-Lys-Lys-OH

Bold charcaters denote modification in the peptide sequence.

**Table 2 microorganisms-10-02386-t002:** Coupling conditions for each step of the peptide synthesis.

Peptide	Cycle															
	1	2	3	4	5	6	7	8	9	10	11	12	13	14	15	16
CSP1	A	A	A + B	A + B	A + A + B + C	A + A	A + A	A + A	A + A	A + A + B + C	A + A	A + A	A + A	A + A	A + A + B + C	A + A + C
AcCSP1	A	A	A + B	A + B	A + A + B + C	A + A	A + A	A + A	A + A	A + A + B + C	A + A	A + A	A + A	A + A	A + A + B + C	A + A + C
Ac4CSP1	A	A	A + B	A + B	A + A + B + C	A + A	A + A	A + A	A + A	A + A + B + C	A + A	A + A	A + A	A + A	A + A + B + C	A + A + C
CSP1hβE	D	D	D	D + C	E + C	E	E	E	E	C	E	E	E	C + E	C	C
CSP1AA	F	F	F	C	C	F	F	F + F	C	C	F	C	C	C	C	C + C
CSP1Y(SO_3_)	D	D	D	D + C	E + C	E	E	E	E	C	E	E	E	C + E	C	C

(A) single coupling with 5eq. AA, 5eq. 0.45M HBTU/HOBt in DMF, 10eq. 2M DIPEA in NMP; (B) single coupling with 5eq. AA, 5eq. 0.45M HATU/HOAt in DMF, 10eq. 2M DIPEA in NMP; (C) single coupling with 8eq. AA, 8eq. 0.95M PyOxym in DMF, 16eq. 2M DIPEA in NMP; (D) single coupling with 5eq. AA, 10eq. 0.95M PyOxym in DMF, 16eq. 2M DIPEA in NMP; (E) single coupling with 10eq. AA, 10eq. 0.95M PyOxym in DMF, 16eq. 2M DIPEA in NMP; (F) single coupling with 5eq. AA, 5eq. 0.95M PyOxym in DMF, 10eq. 2M DIPEA in NMP.

**Table 3 microorganisms-10-02386-t003:** Sequences of the oligonucleotides used in the qRT-PCR experiments.

Gene	Oligonucleotide Sequence	Annealing Temperature (°C)
*gyrA*	fw ^1^ 5′-ATTGTTGCTCGGGCTCTTCCAG-3′ rv ^2^ 5′-ATGCGGCTTGTCAGGAGTAACC-3′	62
*bip*	fw 5′-TGCGGTCTATTGACCTCCTC-3′ rv 5′- CGGGGTACCTTGATTATTTA-3′	60
*vicK*	fw 5′-CACTTTACGCATTCGTTTTGCC-3′ rv 5′-CGTTCTTCTTTTTCCTGTTCGGTC-3′	52
*comC*	fw 5′-CCAAAATGGTATTATGGCTGTCG-3′ rv 5′-TGAGTCTCTATCAAAGTAACGCAG-3′	60
*comD*	fw 5′-GCGCTATTCCTGCAAACTCG-3’ rv 5′-TGACTTGTTTAGGCGGGCAA-3′	60
*comE*	fw 5′-TTCAATGCGGTGGGAGAACT-3′ rv 5′-GGACTGGAAGTAGCCAATCAGA-3′	60

^1^ fw: forward; ^2^ rv: reverse.

**Table 4 microorganisms-10-02386-t004:** Minimum inhibitory concentrations.

Peptide	*S. mutans*	*S. mitis*	*S. oralis*	*S. sanguinis*	*P. aeruginosa*	*F. nucleatum*	*P. gingivalis*
CSP1	30	25	25	21	>50	>50	>50
AcCSP1	20	24	21	27	>50	>50	>50
Ac4CSP1	26	28	23	25	>50	>50	>50
CSP1hβE	20	20	20	20	>50	>50	>50
CSP1AA	27	29	29	27	>50	>50	>50
CSP1Y(SO_3_)	20	20	20	20	>50	>50	>50
CHX	0.63	0.50	0.57	0.69	1.68	300	254

Minimum inhibitory concentrations (MIC) were determined in different bacterial strains using the broth micro-dilution method. Bacteria were incubated for 16 h with peptides (concentration range 0–50 μM) or chlorhexidine (CHX, concentration range 0.05–500 μg/mL) as a positive control under aerobic (*P. aeruginosa*), microaerophilic (*Streptococcus* spp.), or anaerobic (*F. nucleatum* and *P. gingivalis*) conditions. Three separate experiments were performed. Data are reported as μM for peptides and as μg/mL for CHX.

## Data Availability

Not applicable.

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
