# Peer review of "Exploiting Conserved Quorum Sensing Signals in Streptococcus mutans and Streptococcus pneumoniae"

_microorganisms, 2022, doi:10.3390/microorganisms10122386_

Round 1

Reviewer 1 Report

To improve this article, please correct the description as the follows:

1. What is “BlpRH pathway”, which needs introduction about its biological functions? In addition, “higher degree” shown on Line 80 has no specific comparison with others. Thus, it is better written by “high degree”.

2. All bacterial names should be presented in italics. Please check out and correct them through whole manuscript.

3. The source or brand name of agar (line 305) used in this study should be clearly indicated.

4. The analogs of CSP1 are synthesized and used in this study. However, the term of “mutated CSP1 peptides” is confusing because the mutation is commonly used in a biological manner. Please change the term of mutation or mutated through whole article.

5. The description on the front title of Figure 3 is unclear. Please rewrite it for clarity. CSP1 induces bacteriocin production of S. mutans and then inhibits other bacterial growth through bacteriocidal activity. Certainly, CSP1 analogs may also play a similar role in inhibiting bacterial growth as CSP1. Therefore, the description of the results of growth inhibition should not just come from CSP1, but also from the interaction between synthesized CSP1 analogs and CSP1 in a manner of competition.

6. The description on the front title of Figure 4 is also unclear as my explanation for Figure 3 above. Please rewrite it for clarity. 

Author Response

  1. What is “BlpRH pathway”, which needs… Thus, it is better written by “high degree”.

We thank the Reviewer for his/her suggestions. The BlpRH pathway was discussed in more detail in the introduction, and the mistake was corrected.

  1. All bacterial names should be… correct them through whole manuscript

We checked the Manuscript according to the Reviewer’s suggestions. All the names of bacteria are now reported in italics.

  1. The source or brand name of agar (line 305) used in this study should be clearly indicated.

We added the information to the Revised version of the Manuscript.

  1. The analogs of CSP1 are synthesized … Please change the term of mutation or mutated through whole article

The Reviewer is right. The term “mutated“ was changed to “synthesized” in the revised version of the Manuscript.  

  1. The description on the front title of Figure 3 is unclear. …, but also from the interaction between synthesized CSP1 analogs and CSP1 in a manner of competition.

We thank the Reviewer for these observations. We completely agree that the effects we observed in bacteriocin production can be mediated by the competition among different signals. We added a comment about this in the Manuscript. Following the Reviewer’s suggestions, we will perform HPLC analysis on the bacterial conditioned media and checkerboard test to assess the competition.

The front title of Figure 3 was rewritten.  

  1. The description on the front title of Figure 4 ... Please rewrite it for clarity.

Done

Reviewer 2 Report

Dear authors

I find your work valuable and a contribution to the understanding of the understanding of diverse nature of the oral microbiome. 

Author Response

We thank the Reviewer for his/her comment on our Manuscript.

Reviewer 3 Report

Bernabè et al. have done a nice piece of work and it is an interesting to know the Quorum Sensing Signals in pathogenic bacteria. However, the current study needs some minor revisions before the acceptance.

1.     Overall, the manuscript was well written. However, it will be more eye catching and understandable to reader if the authors can add an infographic illustration of the study as it has many experimental setups and outcomes.

2.     Please add a strong conclusive finding to the abstract.

3.     Please add the some more information/definition about biofilms, quorum sensing, inter species/ inter kingdom cross talks.

4.     How can your findings help in futuristic treatment/therapeutic approaches.

5.     Mention the limitations of current study.

Author Response

  1. Overall, the manuscript was well written. However, … as it has many experimental setups and outcomes.

We thank the Reviewer for his/her advice. We generated an infographic image summing up the main experimental approaches and results. The image is reported in Figure 5 of the revised version of the Manuscript. 

  1. Please add a strong conclusive finding to the abstract.

The last sentence of the abstract was reformulated.

  1. Please add the some more information/definition about biofilms, quorum sensing, inter species/ inter kingdom cross talks.

More details about biofilms, quorum sensing, and inter-species and inter-kingdom crosstalk were reported in the Introduction of the revised version of the Manuscript (lines 58-69).

  1. How can your findings help in futuristic treatment/therapeutic approaches.

The results reported in our study identify the functional overlapping of QS-mediated signals between two bacterial species (S. mutans and S. pneumoniae) sharing common signaling pathways and similar autoinducers. As reported in the Discussion section, the higher flexibility of shared QS systems could represent a new target in the interventional approaches that exploit the fratricide interaction among bacterial species growing in similar environmental niches. These observations could benefit both the clinical field and the biotechnological approaches where mixed bacterial cultures are proposed for substrate generation, bioremediation, or muti-steps processing.     

  1. Mention the limitations of current study.

We thank the Review for this suggestion. In the revised version of the Manuscript (lines 447-448), we reported that co-cultures among bacterial species need to be better analyzed as the inhibition zone could be determined by specie competition. Moreover, as reported in the Discussion section (line 583), CSP1 analogs could differently impact the behaviour of commensal species. We will assess these issues in mixed dynamic biofilm mimicking the development of dental caries.